# Analysis of the Hanging Actions and Operating Heights of Storage Furniture Suitable for the Elderly

**DOI:** 10.3390/s23083850

**Published:** 2023-04-10

**Authors:** Xinao Shi, Fan Zhang

**Affiliations:** College of Materials Science and Technology, Beijing Forestry University, Beijing 100083, China

**Keywords:** the elderly, storage furniture, hanging action, operating height, body height, comfort

## Abstract

The current functional scale design of storage furniture which the elderly use does not meet their needs, and unsuitable storage furniture may bring many physiological and psychological problems to their daily lives. The purpose of this study is to start with the hanging operation, to study the factors influencing the hanging operation heights of elderly people undergoing self-care in a standing posture and to determine the research methods to be used to study the appropriate hanging operation height of the elderly so as to provide data and theoretical support for the functional design scale of storage furniture suitable for the elderly. This study quantifies the situations of elderly people’s hanging operation through an sEMG test performed on 18 elderly people at different hanging heights combined with a subjective evaluation before and after the operation and a curve fitting between the integrated sEMG indexes and the test heights. The test results show that the height of the elderly subjects had a significant effect on the hanging operation, and the main power muscles of the suspension operation were the anterior deltoid, upper trapezius and brachioradialis. Elderly people in different height groups had their own performance of the most comfortable hanging operation ranges. The suitable range for the hanging operation was 1536–1728 mm for seniors aged 60 or above whose heights were within the range of 1500–1799 mm, which could obtain a better action view and ensure the comfort of the operation. External hanging products, such as wardrobe hangers and hanging hooks, could also be determined according to this result.

## 1. Introduction

Aging is a natural process of progressive decline in muscle mass, strength [1] and physical function [2,3]. The decline in muscle strength precedes the decline in muscle mass and leads to a decrease in the activities of daily living [4,5], and most elderly people spend more time indoors than outdoors. In terms of the choice of a senior care model, 90% of Chinese older adults prefer home-based care and spend more time in their homes. As a result, it is particularly important to make the living environment of elderly people as convenient as possible.

Storage space occupies a large part of the entire home environment, and storage behaviors in the daily lives of seniors occur with high frequency and appear in all the spaces of the living environment. The decline in muscle capacity raises the problems of reduced operational accessibility and a limited ability to perform daily activities. Compared to younger ages, the range of accessibility is reduced due to a gradual decrease in the human scale, such as height and the highest areas that elderly people can raise their hands to reach, which limits their access to objects at high places in daily life. The decrease in muscle strength also causes discomfort in the lower back and legs when bending and squatting in a lower position. In the long-term repetitive daily storage process, if the elderly need to regularly perform improper storage movements, such as ascending, squatting, bending and standing on their tiptoes, which are beyond their physical performance, they may suffer from soreness, dizziness, the physical inability to stand or even fall due to their lack of physical ability, which is a great safety hazard in daily life [6]. It may also cause or aggravate muscle diseases in elderly people [7] and make their muscle mobility even weaker. At the same time, due to a mismatch between the storage products they use and their needs, elderly people always pile things up everywhere according to their living habits, which causes more problems in their lives because of the chaos in the space and the process of searching for things, which results in more psychological anxiety and powerlessness in their lives.

Although the Chinese elderly industry and the elderly-friendly furniture industry have been developing and improving along with the aging process in order to create a more convenient, comfortable and safe living environment for the elderly, most of the current furniture designs for elderly people are still based on experience, and the design scale of storage furniture used by the elderly is not matched with their operational behaviors after scientific user research. Therefore, the design of age-appropriate furniture still cannot meet the actual needs of Chinese elderly people [8,9,10]. Some researchers have identified these problems and are trying to study the elderly and furniture suitable for them with qualitative and quantitative methods [11,12], but there are still few objective and scientific bases for quantifying the usage behaviors of the elderly and their comfort and for determining the suitable storage operation height for the scale design of storage furniture. User research is an important guide to the transformation of the furniture design language. Matching daily storage behavior with the physical scale and mobility of the elderly is helpful for assisting the elderly in their lives to a certain extent, for completing their daily activities and for meeting the needs of physiological activities while reducing the psychological loss and anxiety of unavailability. For this reason, in order to meet the needs of the elderly for storage furniture, it is still necessary to start from the research on elderly users and their behaviors and to transform the data from elderly storage behavior research into the appropriate design scale for elderly-friendly furniture.

Feasible research instruments can be found from the operational motions of the elderly so as to conduct a systematic and scientific study on the storage behaviors and needs of the elderly. There are currently several experimental tools that can be applied in studies related to operational movements [13]. Many scholars have chosen surface electromyography (sEMG) as a method that is not harmful to subjects and that allows for testing and studying their muscle activities [14,15,16,17]. The current sEMG studies have been conducted by scholars from different disciplines, targeting biomedicine [18], kinesiology [19], wearable devices [20], mobility assessment [21], human–robot interaction and machine-learning [22]. The muscle activities of different genders [16,23,24,25], muscle load [26] and muscle fatigue [27] were studied in in-depth studies on different test vectors. Kim et al. [28] conducted a study to assess individual shoulder muscle fatigue patterns under various external conditions using sEMG. There were also corresponding research advances for different body parts [29,30]. In behavioral studies on elderly people with the application of sEMG, the attachments, intelligent operations and working abilities and physical conditions of different age groups were studied, such as the “Sit-Stand-Sit” process of the safe seat for geriatric patients [31], a comparison of young and elderly people going from sitting to standing in daily life [32] and the hand postures of elderly people [33]. Qin et al. [34] studied the effect of the development of shoulder muscle fatigue during repetitive manual tasks in younger (with a mean age of 25.2) and older women (with a mean age of 61.7) on the activity of the trapezius, anterior middle and posterior deltoid and infraspinatus muscles. Badawy et al. [35], on the other hand, made some comparisons between the activities of the trunk muscles during one-handed lifting in elderly and obese individuals. The activities of the trunk muscles during single-handed lifting in older adults with different BMIs and weight bearing conditions were studied. The existing studies show that it is feasible and non-invasive to apply sEMG to the study of the daily living behaviors of the elderly. However, there is still a gap in the study of the daily living behaviors of the elderly, especially in the sEMG studies of a certain furniture using behaviors and elderly people’s muscle activities. By applying sEMG to analyze the muscle activities in the operating behavior of the elderly when using storage furniture, it can provide objective feedback on the physiological and psychological performance of the elderly in the process of its use, can derive the matching relationship between the design scale of storage furniture and the needs of the elderly and can provide support for the design of the functional scale of storage furniture suitable for the elderly.

In this study, the common storage behaviors of elderly people in their homes were split, involving opening and closing doors, hanging, stacking (or placing directly on the plane) and pushing and pulling drawers. A wardrobe, with a large volume and covering different storage operation behaviors, was taken as the research carrier of this study, and the hanging operation was chosen as the starting point of the basic research on the design scale of storage furniture. Elderly people who were involved in self-care at home were selected as the test subjects for the hanging operation test. In order to determine the optimal operating height range and to avoid the squatting, bending or tiptoeing actions of the elderly, the main muscles involved in the hanging operation in the standing position were selected. The sEMG indexes of the right pectoralis major (PM), biceps brachii (BB), anterior deltoid (AD), triceps brachii (TB), brachioradialis (BR) and upper trapezius (UT) were tested [36,37,38,39]. The human scales of the elderly were introduced in this study, and the subjects were divided into groups according to their heights in the subsequent analysis. The sEMG time and frequency domain indicators were analyzed. Additionally, in order to verify and predict the muscle activities of the subjects at different operation heights, a principal component analysis (PCA) was used to obtain the comprehensive sEMG indexes of the test muscles, and the fitted curves of the comprehensive indexes and test heights were carried out to obtain the lowest sEMG value position, which represented the position with the lowest muscle force and a relatively effort-saving operation. The Likert scale was used to rate the expectation and subjective evaluation of the hanging actions before and after the operation. The objective and subjective methods were combined to study the muscle activities of the elderly during the hanging operation so as to quantify the hanging movements of the elderly in different groups and to obtain the height ranges of the hanging operation suitable for old adults. The results provided useful scientific method and data references for the design and development of the design scales of elderly-friendly wardrobes, clothes rails, hanging hooks and other furniture and products and also improved the comfort and convenience of the elderly during the hanging operation [40,41]. They could also help to create a healthier and more comfortable home atmosphere, thus effectively improving the quality of life of the elderly [6].

## 2. Materials and Methods

### 2.1. Subjects

Eighteen subjects volunteered for this study. The age range of the participants was 63–77 years (mean = 69.94, SD = 4.22). Their mean height and weight were 1688 mm (SD = 74 mm) and 66.75 kg (SD = 9.54 kg), respectively. All subjects were right-handed and had no history of upper extremity injuries or surgeries, wounds, or skin allergies. Table 1 lists the basic information and body scale data of the participants.

In the experiment, gender did not show significant difference (*p* > 0.05), while the height difference was significant (*p* < 0.05). Therefore, the influence of gender on the hanging operation was not considered in the analysis of this paper. Using the minimum 64 mm cell interval of the test’s independent variable as a reference, the subjects were equally divided into three height groups of 1500–1599 mm (noted as G150), 1600–1699 mm (noted as G160) and 1700–1799 mm (noted as G170) for analysis.

### 2.2. Wardrobe and Clothing Samples

This test simulated the use of furniture in the home environment of the elderly to the greatest extent possible, eliminating their discomfort with the test itself and making the operation more natural and closer to a real-use situation. Therefore, a wardrobe model (Figure 1a) with appropriate scale and adjustable hanging height was required while excluding the influence of appearance, color and material on the test results. A wardrobe frame with dimensions of 750 × 580 × 2360 mm and a clothes rail with dimensions of 708 × 25 mm were selected as the test model. The clothes rail was fixed with flanges (48 × 19 × 3 mm) and could be adjusted in height, with adjustable intervals referring to the 32 mm system for panel furniture which was still commonly used in the standardization of furniture structure design. Every 64 mm was as an adjustable height.

A long down coat, which was relevant to the hanging movements in daily life, was selected as the clothing model. The hanging length (from the lower edge of the hanger hook to the lower edge of the garment) of the test clothing was about 1200 mm, and the weight was about 1250 g. The hanger used for hanging the down coat was a common hanger for adults in the market. The test clothing was covered with a clothing dust cover to prevent irregular deformation during the test operation from affecting the test results. The overall model of the test clothing is shown in Figure 1b.

### 2.3. Instrumentation

Surface electromyography (sEMG) signals were used to quantify muscle activities at different test heights. An 8-channel sEMG signal measurement and acquisition system (Kingfar Technology Inc., Beijing, China) was used to record and process the sEMG signals. The sEMG signals were sampled at a frequency of 1000 Hz with a band-pass filter tuned at 10–500 Hz. The wireless EMG sensors were used for sEMG signal acquisition, and the signal was transmitted to ErgoLAB human–machine environment synchronous test physiology cloud platform for data pre-processing and sEMG feature extraction.

### 2.4. Experimental Design and Procedure

#### 2.4.1. Experimental Design

In this test, the hanging rail height of the wardrobe was set to 8 levels, which were 1536 mm, 1600 mm, 1664 mm, 1728 mm, 1792 mm, 1856 mm, 1920 mm and 1984 mm. Every 64 mm was a test interval. The heights of the hanging rail were set according to the test hanging action and the human scales of the elderly so as to ensure that the operation movements were generally consistent during the test and to avoid squatting, bending or tiptoeing of the elderly as far as possible. The test heights were combined with the human scales of Chinese elderly people [42] to ensure that they could accurately see the hanging position at the lower height and could reach the higher position. The heights of the hanging rail were set from the average sight height of the elderly to around their hand function height, corresponding to the design scale of the wardrobe in this experiment, namely, 1536–1984 mm.

As for the selection of test muscles, since the test movement was one-handed and since all subjects were right-handed, the main muscles tested in this experiment were the right pectoralis major (PM), biceps brachii (BB), anterior deltoid (AD), triceps brachii (TB), brachioradialis (BR) and upper trapezius (UT), which played important functional roles in the hanging movement.

The total duration of the participants’ test was less than 1 h to prevent physical and psychological discomfort of elderly subjects due to the experiment.

#### 2.4.2. Experimental Procedure

The procedure of this test was divided into preparation, MVC measurement, formal test operation and subjective evaluation.

(1)Preparation

Prior to the start of the experiment, each participant signed a written voluntary agreement giving a brief overview of the study procedure to ensure that every subject was fully informed and consented to the experiment. The basic information of the subjects was recorded, and their heights, their weights and other functional heights related to the hanging operation were measured as the basis for subsequent analysis.

Before applying the electrodes, the skin of the electrode application site was thoroughly cleaned with medical scrub and medical alcohol [43]. Disposable bipolar Ag/AgCl sEMG electrodes with a diameter of 50 mm were applied to the protrusions of the six tested muscles [44], and the two electrodes centered 25 mm apart in diameter [45]. The electrodes were attached to the middle of the muscles and parallel to them. The positions of the electrodes were as shown in Figure 2 and Table 2. The temperature of the test environment was constant and suitable.

(2)MVC measurement

MVC measurements were performed on the six muscles mentioned above [44]. The tested muscles intermittently contracted 3 times, and they contracted for 3 s and rested for 20 s. The obtained MVC values were used to normalize the data for subsequent tests. Subjects had 10 min of rest after each muscle MVC test until each muscle potential was fully recovered, ensuring that the subject’s target muscle was fully rested before the formal test [46,47,48].

(3)Formal test operation

The formal test operation was the same as MVC test preparation, starting with affixing the subject’s target muscle electrode and adjusting the equipment for formal EMG signal data measurement. The subject stood at a relative position 400 mm away from the hanging position, which simulated the actual using distance in daily life and was not affected by the installation position of the clothes rail in the horizontal direction. The clothes rail was positioned at the height of 1536 mm at first. Before the test started, the subject stood with their hands hanging naturally on both sides of their body. After the “ready” command, the subject’s right forearm was naturally bent and perpendicular to their upper arm, and their upper arm was close to the side of their body. After the “start” command, the subject held the test clothing handed by the staff, hung it on the clothes rail within 5 s in the customary position and lowered their right arm to its natural hanging position for one operation. The above process was repeated three times for each test height, with a 20 s break between each operation to restore the tested muscles’ states. The rest time between each test height was 2 min. The operating action of each test height is shown in Figure 3. The sEMG signal was simultaneously recorded and transmitted to the data processing platform along with the test movement to obtain the raw sEMG data and signal images for subsequent processing.

(4)Subjective evaluation

In order to obtain the subjective evaluation of the overall operating comfort of each test height to supplement the results of the sEMG test, the expectation score and the subjective evaluation score after operating were calculated for each test height before the start of the operation and during the rest time after the completion of the operation. The expectation scores were based on the subjects’ visual senses, usage habits and experience, while the subjective evaluation scores after operating were based on the experience of the hanging test. The rating was based on the Likert scale, which set the comfort level into 7 levels (−3 represented the least comfortable, 0 represented no effect, and 3 represented the most comfortable). The subjective evaluation of the comfort level is shown in Table 3. The subjective evaluations were recorded by the staff. Once all of the above operations were completed at 8 test heights, the test was completed for the subject. The operation process at one test height is shown in Figure 4.

### 2.5. Measurement and Data Processing of sEMG Signals

sEMG signals were obtained at 1000 Hz (sampling rate), and their bandwidth filters ranged from 10 to 500 Hz. The white noise was removed from sEMG signals by using wavelet denoising. Root mean square (RMS) values of raw sEMG data were calculated with moving window of 100 ms, and their means were quantified. In this study, time domain analysis and frequency domain analysis were chosen for sEMG signal feature extraction. Time domain analysis is a method used to calculate the characteristics of the time–potential curve of the filtered EMG signal, which could characterize the amplitude of the EMG signal. Frequency domain analysis was used to perform Fast Fourier Transformation (FFT) on the sEMG signal to obtain the frequency spectrum or power spectrum curve of the signal and to quantitatively respond to the variation characteristics of sEMG through the changes in the signal in different frequency dimensions [49,50]. In this study, RMS (root mean square) was chosen to estimate the magnitude of muscle-generated force and to evaluate whether muscle fatigue occurred during exercise: the larger the value, the more the muscle tended to be fatigued [51,52,53]. iEMG (integrated electromyography) is the area of integration of the amplitude under an sEMG signal at a certain time, responding to the overall strength of muscle activities, and can be used to determine the contribution of each muscle under the same operating action. MF (media frequency) was used to analyze the degree of muscle fatigue generation. With exercise, if muscle fatigue occurred, MF shifted from high to low frequencies as indicated by a gradual decrease in MF [54,55]. A negative slope could indicate muscle fatigue, with a more negative slope indicating greater muscle fatigue [56,57,58,59].

The formulas of RMS (1), iEMG (2) and MF (3) are as follows [52]:(1)$$\mathrm{RMS}=\sqrt{\frac{1}{N}\int_{i=1}^{N}\mathrm{EMG}{\left(i\right)}^{2}}$$
(2)$$\mathrm{iEMG}=\int_{N2}^{N1}X\left(t\right)\mathrm{dt}\;$$
(3)$$\overset{\mathrm{MDF}}{\underset{0}{\int }}P(f)\mathrm{df}=\int_{\mathrm{MDF}\;}^{\infty }P(f)\mathrm{df}$$

The formula of muscle contribution rate calculated through iEMG is as follows [19]:$$\eta =(I_{n}/\left(I_{1}+I_{2}+I_{3}+\cdots +I_{n}\right))\times 100\%\;\left(n=1,2,3,4,5\right)$$

### 2.6. Statistical Analysis

Statistical analysis was performed by applying the statistical software SPSS 26.0 (IBM Corp., Armonk, NY, USA). One-way ANOVA was used to analyze the effects of subject height, different muscles and different test heights on muscle activity and subjective evaluation of the subjects under hanging operation. Pearson correlation coefficient was used to analyze the relationship between test height, sEMG signal, subject height and subjective evaluation. The α level was set at 0.05, which was statistically significant.

## 3. Results

The correlation between the sEMG data and the test heights is shown in Table 4. Among them, except for the MF value of the AD, which was not significantly correlated with the test heights (*p* > 0.05), all the other test indexes were significantly correlated with the test height (*p* < 0.05), so the test design and parameter selection were reasonable and feasible for the follow-up analysis.

### 3.1. Analysis of Contribution Rates of Muscles

The muscle contribution rate of the subjects at different test heights could be used to determine the main power muscles at a certain test height for subsequent primary analysis and, to a certain extent, to reflect the changes in the muscle activities and operating postures. The statistical analysis, through an ANOVA, showed significant differences in iEMG between the subjects (F = 20.963, *p* = 0.000). The muscle contributions of the elderly subjects at different test heights were calculated separately as shown in Figure 5. It could be seen that the main power muscles of the elderly subjects during the hanging operation were the AD, UT and BR, which would also be used as the main muscles for the subsequent sEMG analysis of the muscle activities.

### 3.2. Muscle Activities at Different Test Heights

The ANOVA of the different muscles under the same group yielded significant differences (*p* < 0.05) in their sEMG values, allowing for a subsequent discussion of the activity of the different muscles. The significance of the differences is shown in Table 5.

#### 3.2.1. Activities of the AD

The ANOVA of the sEMG indexes of the AD showed that there were significant differences in the RMS (F = 7.399, *p* = 0.004) and the MF (F = 115.561, *p* = 0.000) values between the different groups, with significant differences (*p* < 0.05) between G150 and G160 as well as G150 and G170. There was no significant difference between G160 and G170. The MF values were significantly different among all three groups (*p* < 0.05), and the significance of the differences is shown in Table 6.

The activity of the AD at different test heights is shown in Figure 6a. From the overall force situation, the force of the AD gradually decreased as the height of the subjects increased, and G150 was the group with the greatest force during the whole process. As the height of the test increased, the force of the AD gradually increased in all three groups. The MF of G150 first rose and then gradually showed a downward trend after 1728 mm, but the overall downward trend was relatively stable. The MF of G160 showed a downward trend at 1536–1664 mm and 1728–1984 mm in which the downward trend was more obvious in the range of 1536–1664 mm. Combined with the rise in the RMS, the AD gradually accumulated muscle fatigue. The MF values of G170 showed an increasing trend during the whole process, and the muscle force gradually increased with the increase in the test height but did not reach the degree of fatigue.

#### 3.2.2. Activities of the UT

The different groups had significant effects on the RMS (F = 9.668, *p* = 0.001) and the MF (F = 5.148, *p* = 0.015) of the UT as shown in Table 6. The RMS of all three groups were significantly different (*p* < 0.05), while only the MF of G150 and G170 had significant differences (*p* < 0.05).

The activity of the UT at different test heights is shown in Figure 6b. The UT activity was still the highest in the group G150 followed by G160 and G170, and the UT activity tended to increase in each group as the test height gradually increased. In combination with the MF values, the MF of all three groups increased more steadily, so there was no accumulation of fatigue during the whole process.

#### 3.2.3. Activities of the BR

The groups had a significant effect on both the RMS (F = 73.825, *p* = 0.000) and the MF (F = 9.363, *p* = 0.001) of the BR. All three groups had significant differences in their RMS (*p* < 0.05), while G150 and G170 did not have significant differences in their MF as shown in Table 6.

The activity of the BR at different test heights is shown in Figure 6c. G150 was the group with the greatest force, gradually decreasing from 1536 to 1664 mm, reaching the lowest point of force at 1664 mm and then gradually increasing. G160 rose, decreased from 1536 mm to 1664 mm and then gradually increased with the different heights. The lowest point of force was at 1664 mm. G170 had a more stable overall change, decreasing and then increasing in stability, and the minimum force was at 1664 mm. In terms of MF, G150 showed a decreasing trend at 1536–1600 mm and 1728–1920 mm, respectively, while the RMS values gradually increased from 1728 to 1920 mm, thus gradually producing some fatigue in this range. G160 and G170 showed an increasing trend throughout the overall process, with a slight decreasing trend at 1664–1728 mm.

#### 3.2.4. Comprehensive Analysis of sEMG Indexes

In addition to the main power muscles, the other three tested muscles also had some influence on the hanging operation, and the force situations are shown in Figure 6d–f. The force of the PM in the three groups was not regular. The PM of G150 reached its minimum force at 1792 mm and then gradually increased, and the MF showed an overall decreasing trend. As a result, the PM of G150 gradually tended to be fatigued after 1792 mm. The PM of G160 changed more smoothly. There was an obvious increase after 1792 mm, and the MF had an overall declining trend from 1536 mm to 1856 mm in which the decline was more moderate between 1600 mm and 1792 mm. The RMS of G170′s PM gradually increased to 1920 mm with the increase in the test heights, while the MF value fluctuated more and did not have a more obvious fatigue trend.

For the BB, G150 had the greatest overall force out of the three groups. G150 had a gradual increase in RMS and an overall increase in MF with the increase in the test height. G160 and G170 both had the smallest RMS at 1664 mm and then gradually increased. The MF values of G160 also showed an overall increasing trend, while G170 showed a gradual decrease after 1728 mm, with a more obvious decrease from 1728 to 1856 mm.

In terms of the TB, G170 still used the least power out of the three groups, with an overall upward trend in the RMS and MF. The RMS of both G150 and G160 increased with the test height. The MF of G150 also showed an overall upward trend, but the MF of G160 decreased significantly in the range of 1728–1856 mm.

According to the comprehensive view of the different muscles’ activities in the range of the test heights, G150 was more comfortable from 1536 mm to 1728 mm, while G160′s comfortable range was 1600–1728 mm. Additionally, G170 was more comfortable from 1600 mm to 1856 mm.

### 3.3. Results of Subjective Evaluation

In addition to the objective test results, the expected scores combined with the usage habits and the use of existing household products as well as the subjective evaluations after operating could, to a certain extent, supplement the errors and problems of the test results. The expected ratings and subjective evaluation scores of the hanging operation from the subjects of the different groups are shown in Figure 7. There were significant differences in the subjective evaluation scores among the groups (F = 8.358, *p* = 0.004), with G150 being significantly different from G160 (*p* = 0.011) and G170 (*p* = 0.001), respectively. G160 was not significantly different from G170 nevertheless.

For the different groups of elderly subjects, the overall expected scores were lower than the actual operational scores. G150 had lower overall scores, and the expected scores were higher than the actual operational scores in the height range of 1856–1984 mm. However, some inconvenience was found after the test operation. G170 had a higher overall rating and a better operation feeling due to the subjects’ heights, and the evaluation scores reached their peak at 1792 mm.

The subjective evaluation box plot of the subjects after the operation is shown in Figure 8. According to the scoring standard, the range of zero and above, which was the subjective feeling of no effect to the most comfortable range, was the optimal scoring range for the comfort evaluation. The subjective comfort intervals of the different groups were as follows: that of G150 was 1600–1664 mm, that of G160 was 1536–1728 mm and that of G170 was 1600–1856 mm.

### 3.4. Fitting Models of sEMG Indicators and Test Heights

The above analysis was based on the analysis of the sEMG indicators iEMG, RMS and MF as well as the subjective evaluation, which corresponded to the height range formed by the eight height points set up for the test. In order to comprehensively analyze the muscle force in the test height range of 1536–1984 mm and to predict the overall muscle force in that height range, which was not tested in the test, a curve fitting was applied to the sEMG indexes and the test heights.

In order to obtain fitting models of the sEMG indexes and test heights and to comprehensively demonstrate the muscle activities of the hanging operation, this experiment applied a principal component analysis (PCA) to synthesize and simplify the RMS of the six tested muscles to fit them with the test heights. The correlation matrix among the different test muscles is shown in Table 7, and the correlation between the different test muscle indexes was suitable for the PCA.

The principal components were extracted according to the principle that the cumulative contribution rate of the PCA was greater than 85%. The variance percentage and cumulative percentage results of the principal component characteristic values are shown in Table 8. It could be seen from the table that the four new indicators contained more than 89% of the information of the original indicators. According to the scoring coefficient and calculation formula of the PCA, the new indexes and comprehensive index could be obtained. The calculation formulas of the new indexes Y1, Y2, Y3 and Y4 and the comprehensive index Y are as follows:$$Y1=F1\times \sqrt{\lambda 1}$$
$$Y2=F2\times \sqrt{\lambda 2}$$
$$Y3=F3\times \sqrt{\lambda 3}$$
$$Y4=F4\times \sqrt{\lambda 4}$$
Y=Y1×δ1%+Y2×δ2%+Y3×δ3%+Y4×δ4%

Note that λ represents the square root of the eigenvalue and that δ% represents the percentage of variance.

The composite index Y was fitted to the test heights, and the fitted curves of the different groups are shown in Figure 9. The formulas of the fitted curves are as follows:


G150
$$Y=14.17-0.02X+8.49\times 10^{-6}X^{2}$$
(R^2^ = 0.96)

G160
$$Y=39.98-0.05X+1.72\times 10^{-5}X^{2}$$
(R^2^ = 0.99)

G170
$$Y=-3.33-0.004X+3.04\times 10^{-6}X^{2}$$
(R^2^ = 0.95)

The trends of the fitted curves of the composite index Y and the test height X of the different groups were all quadratic function curves. In the range of test heights (1536–1984 mm), the X value corresponding to the smallest composite index Y of all three groups was 1536 mm, and the overall muscle force gradually increased with the increase in the test height; however, the growth was different. The slope of G160 changed most significantly among the three groups of fitted curves and was influenced by the change in the test heights, while the slope of the G170 fitted curve changed the least.

## 4. Discussion

Because of the mismatch of storage furniture and elderly people’s daily life needs and the lack of a study method for an elderly-friendly furniture design scale, this study aimed to start from the hanging operation and to explore the influence of hanging operation heights on the muscle activities and subjective perception of the elderly, quantified the hanging operation activities and provided guidance and support for the design of the hanging scale of storage furniture suitable for the elderly. Through the analysis of the sEMG indexes of the muscle activities, subjective evaluations, and results of the fitted curves of the comprehensive sEMG indexes and test heights, the elderly-friendly hanging operation range could be determined, respectively. After the comprehensive analysis, the appropriate hanging operation height ranges of elderly people with different heights were obtained. G150 was more comfortable in the range of 1536–1664 mm. G160 was more comfortable in the range of 1536–1728 mm, and the comfortable hanging range of G170 was 1600–1856 mm.

The results of the sEMG tests showed that the force level of the main power muscles of the different groups during the hanging operation basically remained above 5% MVC, and the force could reach 35% MVC in the case of a higher force with the gradual increase in the test heights. According to the studies on muscle fatigue, it was proposed that, when the muscle load was at a 15–20% MVC level, a decrease in the frequency domain indicators and an increase in the time domain indexes represented the appearance of a fatigue state [27,60,61]. The operation process of this study was short-term with multiple operation movements, and the force levels of the test muscles in the many groups were above 15% MVC. Therefore, this method of judging muscle fatigue could still be used to determine the comfort range of the hanging operation in this test.

The results showed that the subjects’ heights and the test heights had significant effects on the hanging operation behaviors, and the muscle force and operation posture of the subjects in the different groups changed during the gradual increase in the test heights, which caused the differences in the muscle sEMG values and subjective evaluations. Since there was no corresponding standard for the anthropometric measurements of Chinese elderly people, the subject height ranges (1500–1799 mm) were selected to cover the 95th percentile and above of Chinese adults by referring to the “Chinese Adult Anthropometric Measurements” [62], and, also, the mean height of 1672 mm for men and 1549 mm for women were as measured by the Chinese elderly anthropometric measurements in “Housing for the elderly” [42]. The average heights of male and female elderly people mentioned in “A report on the application of anthropometric data of elderly” by Yu et al. [27] were also covered. The test height range was determined to be 1536–1984 mm, which corresponded to the accessible range from the average height of sight to the height of lifting one’s hands up with respect to Chinese elderly people on the human scale [42]. In the three test groups, G150 had three changes in muscle activity over the eight test heights, which were at 1536–1600 mm, 1600–1664 mm and 1856–1920 mm. From the perspective of the muscle activities and subjective evaluations, G160 was most affected by the test heights and the distance between the standing position and the hanging point within the test height range. There were five changes in the muscle activities over the eight test heights, and the subjects adjusted their hanging postures many times to find the appropriate hanging actions. Due to the subjects’ higher heights, G170 was affected the least over the whole operation process. There were two adjustments over the different test heights, respectively, at 1600–1664 mm and 1664–1728 mm. It showed that 1664 mm was a turning point of the hanging heights. Except for a few lower test positions, the muscle power situations of G170 were more similar after 1728 mm.

During the experiment, according to the descriptions of the subjects, the hanging positions of the existing wardrobe products which were used at home were often set by the height of the floor and the top panel of the wardrobe itself. Most of the wardrobe hanging positions of the subjects’ homes were higher than their actual upright positions when raising their hands, so they needed to be used with the help of standing on their tiptoes and other movements, which was not convenient for the elderly users. As a result, for the elderly subjects in the different groups, the expected scores of the subjective evaluations before operating were, overall, lower than the actual operation scores. The overall rating scores of G150 were low, and, in the high operating range of 1856–1984 mm, the expected ratings were higher than the actual operating scores due to the adaptability of the use of existing household furniture. However, they were found to be inconvenient to use after the test operation. The overall trend of G160 was similar to that of G150, with the lowest rating at 1984 mm, but the actual operation scores did not reflect a worse rating than the expected ones. Overall, 1536–1856 mm was a more convenient area for G160 in the hanging operation. G170 scored higher on their overall feedback due to their taller heights, reaching the peak of their operation scores at 1792 mm. During the subjective evaluations, the participants combined their sensations of upper limb operations, waist and leg force and sight in the scores. In a previous study, Badawy et al. [35] proposed that elderly people could occasionally carry loads of up to 10 kg in one hand. In this study, the weight of the hanging objects rarely reached 10 kg or above, and, due to the long-term adaptability of the use and the relatively unclear overall sensory response of the elderly, there were some vague evaluation descriptions and inaccurate scores in the process of subjective evaluation. According to the subjective scores, for the elderly subjects whose height ranges were from 1500 to 1799 mm, the optimal hanging height range, from their subjective perception, was between 1600 and 1664 mm.

Based on the results of the above analysis, the appropriate operating range for the hanging operation was 1536–1728 mm for elderly people over 60 years old with a height range of 1500–1799 mm. The projection distance between the standing position and the hanging point could be adjusted according to the operating habits to achieve a more comfortable operating posture. Yang and Xiong [40] proposed in their analysis of the age-friendly wardrobe that the hanging area could be set in the lower part of the wardrobe, with a height of about 1.5 m, so that the elderly could easily access the clothes. Fang [11] came up with the idea that the height of the clothes rail should be adjusted down to about 1400 mm. The scale design of the hanging position proposed by these two researchers coordinated the overall design proportion of the wardrobe and the design scale of the stacking area, and the overall design height was lower than the relatively comfortable hanging range of this study. Fan [63] mentioned in the research and application of the modular design of the custom wardrobe of Northeast China that the storage scale of hanging clothes was 820–1400 mm, so the internal height of the hanging module of long clothes should be more than 1460 mm considering the distance from the clothes rail to the top panel of the wardrobe. In addition, the height of the wardrobe could be divided into five areas according to the sight ranges of people when taking things, force differences and body movement ranges, respectively: the first area was 600–1500 mm, the second area was 1500–2200 mm, the third area was higher than 2200 mm, the fourth area was 300–600 mm and the fifth area was 0–300 mm. The first area of the wardrobe (600–1500 mm) was the height at which it was the easiest for people to access items in a standing posture, and the second area (1500–2200 mm) was the height at which people could reach from the shoulder position to the position of lifting their arm 60 degrees when standing. In this range, it was possible to pick up items without making too many movements and to have a good operating view. The comfortable hanging range determined in this study was within the second area of the wardrobe design proposed by Fan [63]. The angle from the shoulder position to the arm lifting position was less than 25 degrees, which could not only obtain a good operating sight but could also meet the size requirements of long hanging clothes, proving the scientificity of the test results to a certain extent. In addition to the hanging operation of the wardrobe, the height positions of hanging storage products, such as hanging hooks attached to doors or walls designed for the elderly, could also be determined with a reference to the comfortable operating height range derived from this test.

In this study, the one-handed hanging operation used by the elderly was carried out, and the hanging item involved was not heavy. However, if the item was heavy, one-handed loading would not be the preferred method of loading from a biomechanical perspective. It had also been mentioned in previous studies that a single-handed load would increase the spinal load compared with a two-handed load, even at double the load [64]. Therefore, based on the appropriate range of the one-handed hanging operation obtained in this study, it was suggested that the elderly should avoid one-handed loading when carrying out the hanging operation of heavy objects, if possible, to minimize the activation imbalance of bilateral muscles [65]. The elderly could choose the suitable way to distribute the load to complete the hanging storage operation behavior and to reduce a certain amount of muscle fatigue.

During the test, some errors were made in the test results due to various reasons. First of all, 18 elderly people were selected as the subjects in the experiment, and the number of subjects and the sEMG data of each group decreased after grouping, which would have a certain impact on the results. In addition, when selecting the elderly subjects, although the subjects were limited to healthy elderly individuals undergoing self-care, the physical abilities of the elderly were difficult to keep consistent due to individual differences. There were differences in their physical abilities because of the differences in all the subjects’ work backgrounds, living habits and exercise conditions, which also caused the instability of the test results. Moreover, the elderly had different concentrations, long-term movement habits and individual cooperation conditions. When the test was performed according to the restricted movements, the test results might have been affected by the intentional control of the movements or other movements other than the prescribed ones due to inattention.

In future studies, more elderly subjects could be tested by applying this study process, and elderly subjects with different physical abilities could also be selected. At the same time, tests could also be conducted on the lumbar and leg muscles in addition to the main power upper limb muscles combined with other storage operation postures for a comprehensive study.

## 5. Conclusions

This study focused on the hanging operation and the storage behaviors of the elderly. A total of 18 homecare healthy elderly people (9 females and 9 males) was selected as the experimental subjects to conduct the hanging operation experiment. sEMG was used to test the muscle activities of the right pectoralis major, biceps brachii, anterior deltoid, triceps brachii, brachioradialis and upper trapezius. The subjects were grouped according to their heights (G150, G160 and G170), and the sEMG time and frequency domain indicators were analyzed. The Likert scale was applied to score the subjective evaluations before and after the hanging operation. The comprehensive sEMG indicators of the test muscles were obtained through a PCA, and the fitted curves of the comprehensive indicators and test heights were carried out. It was found that the heights of the subjects, different test muscles and test heights all affected the hanging operation. For the hanging operation, the anterior deltoid, brachioradialis and upper trapezius were the main power muscles of the upper limb. A combination of objective and subjective methods was used to obtain the appropriate operating height range for the different heights of elderly people, i.e., G150 was more comfortable in the range of 1536–1664 mm, 1536–1728 mm was more appropriate for G160, and G170 felt more comfortable in the range of 1600–1856 mm. The combined results showed that the age-appropriate hanging operation range was 1536–1728 mm for the elderly people whose height range was 1500–1799 mm, which could obtain a better operating view and ensure the comfort and convenience of the hanging operation. The projection distance from the stand position to the hanging point could be adjusted according to the operating habits to achieve a more comfortable operating posture. The results provided research method and scientific data support for the scale design of the hanging height of age-friendly storage furniture. The suitable operating height ranges of the hanging operation for elderly people were obtained, and, to a certain extent, this study provided assistance and support for the daily life behaviors and psychological needs of elderly people.

## Figures and Tables

**Figure 1 sensors-23-03850-f001:**
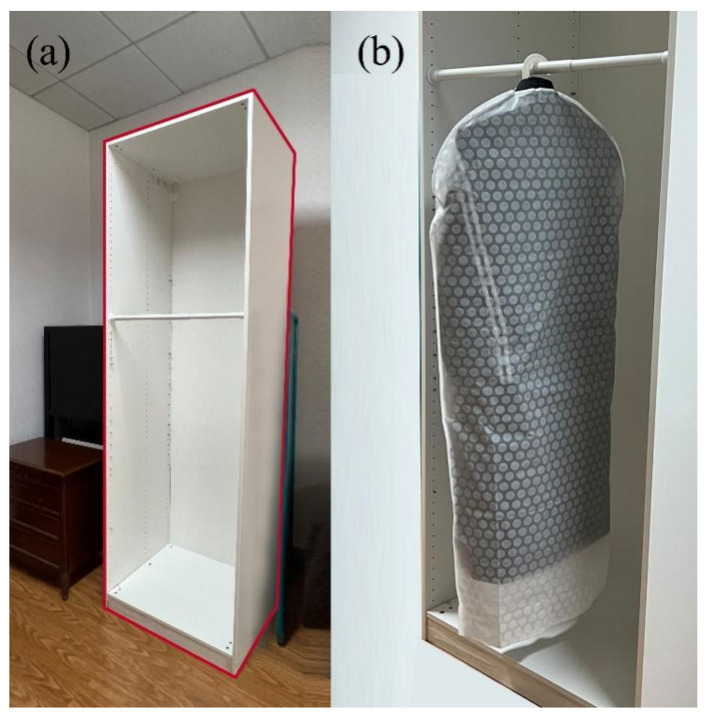
Wardrobe and clothing samples. (**a**) The experimental wardrobe and (**b**) the experimental clothing.

**Figure 2 sensors-23-03850-f002:**
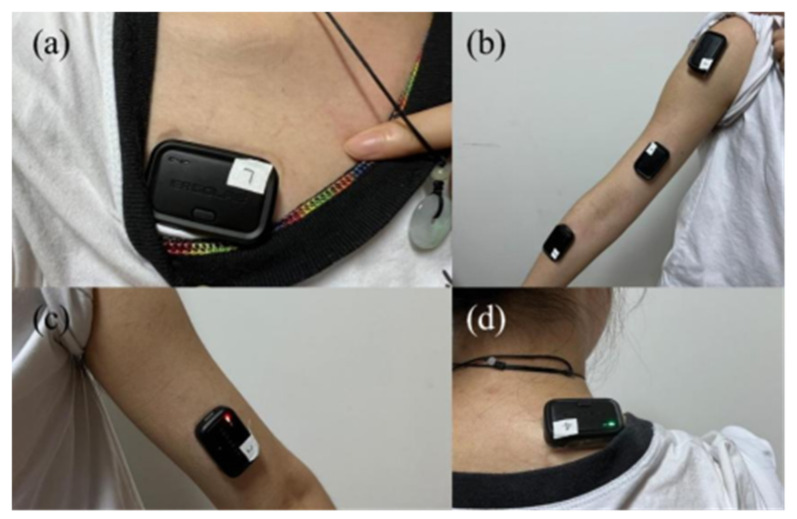
Electrodes’ sticking positions. (**a**) Electrodes’ sticking positions on PM; (**b**) electrodes’ sticking positions on AD, BB and BR; (**c**) electrodes’ sticking positions on TB; and (**d**) electrodes’ sticking positions on UT.

**Figure 3 sensors-23-03850-f003:**
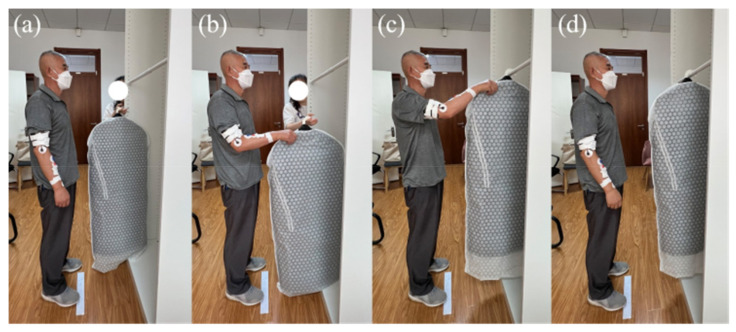
The operation actions. (**a**) The subject stood with their hands hanging naturally on both sides of their body. (**b**) The subject’s right forearm was naturally bent and perpendicular to their upper arm, and their upper arm was close to the side of their body. (**c**) The subject hung the clothing on the clothes rail. (**d**) The subject lowered their right arm to its natural hanging position.

**Figure 4 sensors-23-03850-f004:**
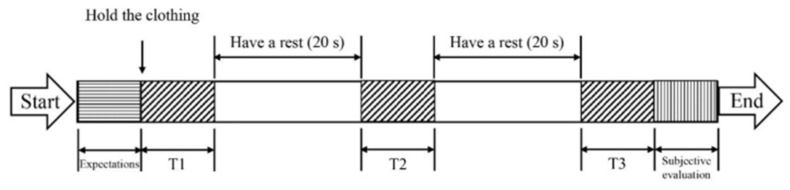
The operation process at one test height. T1: Hang the clothing on the clothes rail and lower the arm into a natural hanging position within 5 s. Note: The operation was the same during T2 and T3.

**Figure 5 sensors-23-03850-f005:**
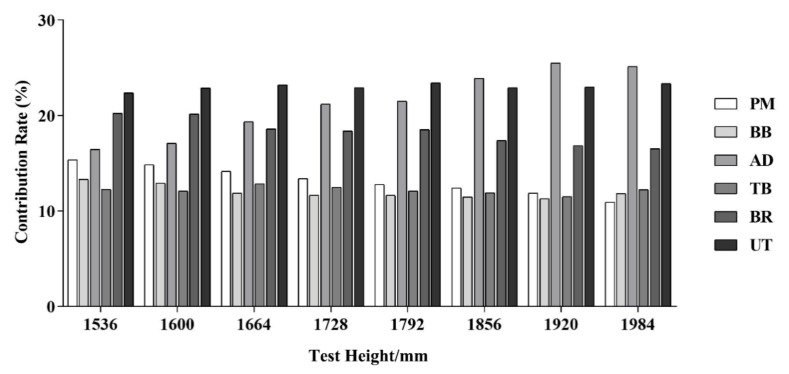
Contribution rates of muscles.

**Figure 6 sensors-23-03850-f006:**
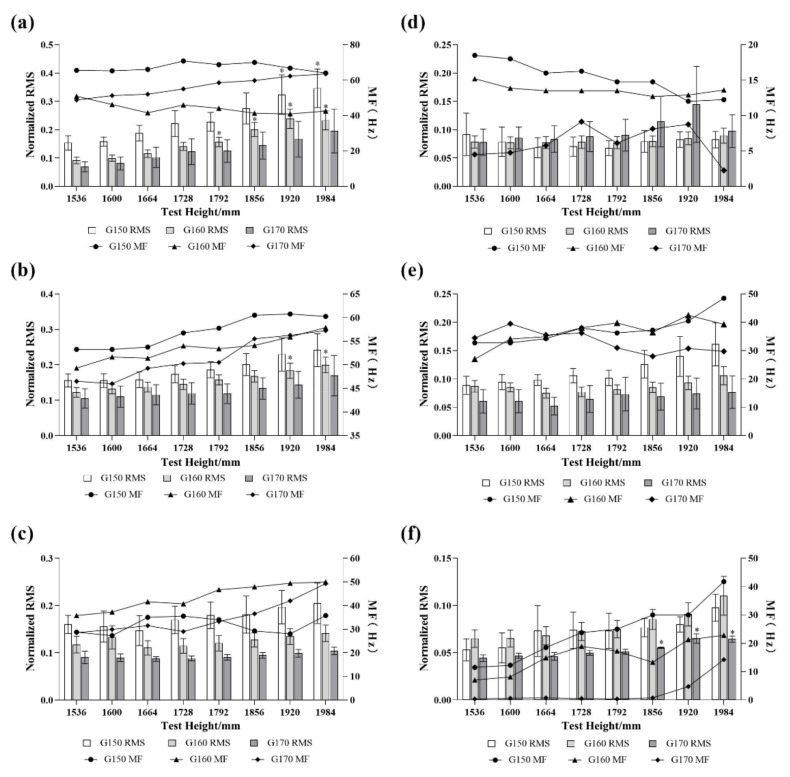
The force situation of different muscles. (**a**) AD, (**b**) UT, (**c**) BR, (**d**) PM, (**e**) BB and (**f**) TB.

**Figure 7 sensors-23-03850-f007:**
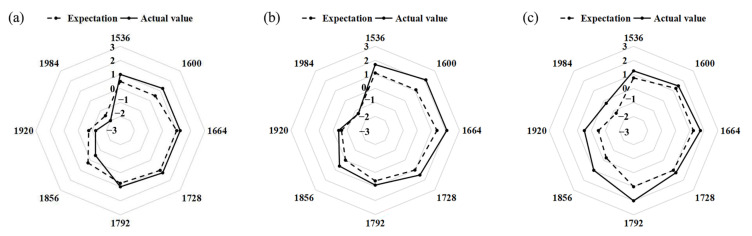
The radar chart of subjective evaluation. (**a**) The subjective evaluation scores of G150 of different test heights. (**b**) The subjective evaluation scores of G160. (**c**) The subjective evaluation scores of G170.

**Figure 8 sensors-23-03850-f008:**
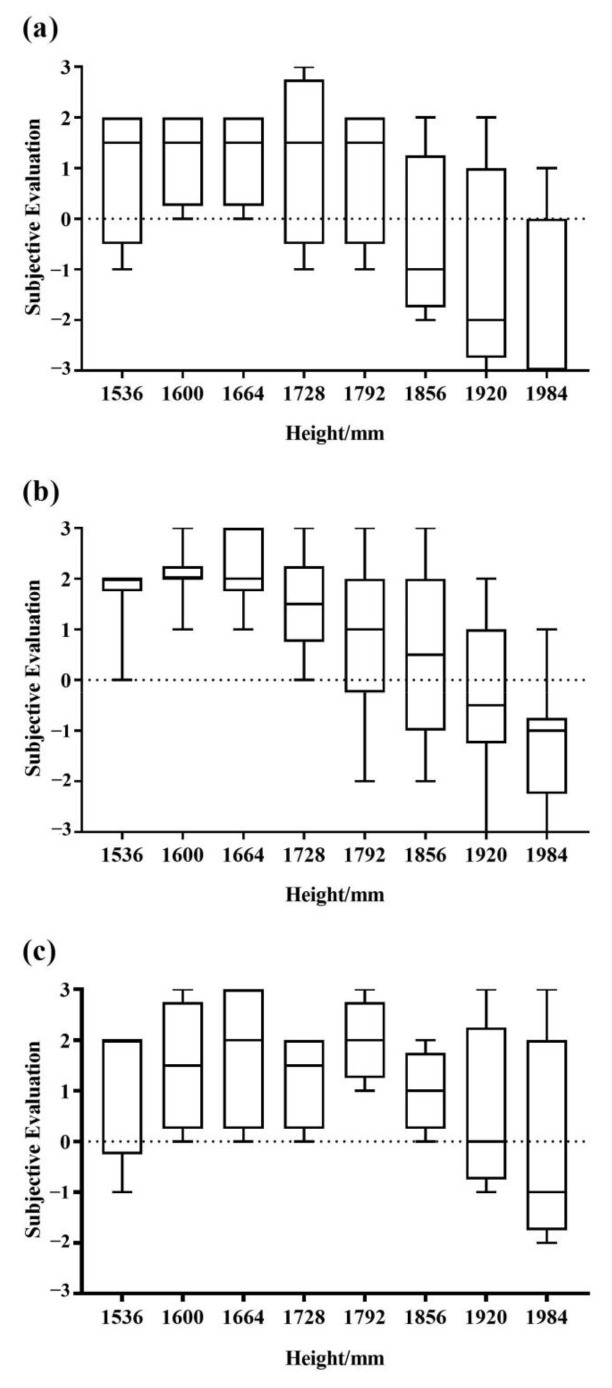
The subjective evaluation after operating. (**a**) Scores of G150, (**b**) scores of G160 and (**c**) scores of G170.

**Figure 9 sensors-23-03850-f009:**
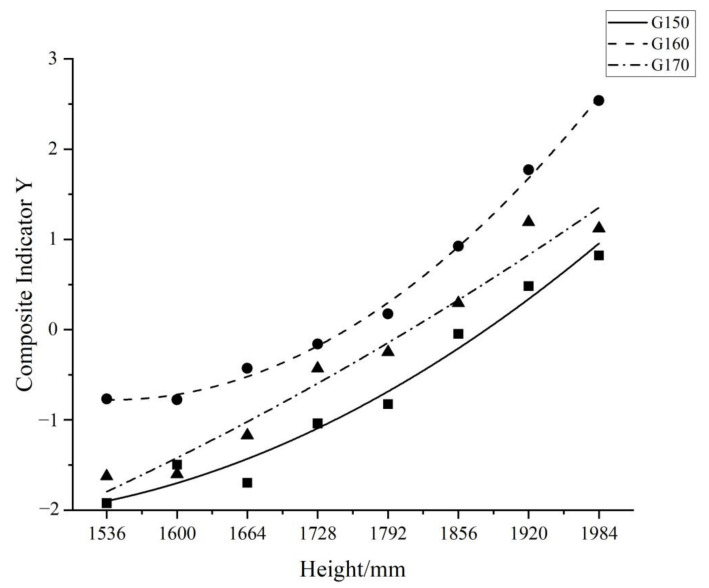
Fitted curves of 3 different subjects’ groups (G150, G160, G170).

**Table 1 sensors-23-03850-t001:** Information of participants’ age and major anthropometric measurements.

Participants	Age (Year)	Height (mm)	Weight (kg)	Eye Level (mm)	Arms Forward Horizontally (Standing) (mm)	Hands Up Straight (Standing) (mm)
All (*n* = 18)	69.94 ± 4.22	1688.06 ± 74.12	66.75 ± 9.54	1565.44 ± 76.56	1316.33 ± 76.32	2025.00 ± 86.48
Male (*n* = 9)	69.78 ± 4.80	1718.44 ± 46.73	70.61 ± 8.99	1614.89 ± 48.15	1374.11 ± 47.22	2069.11 ± 62.05
Female (*n* = 9)	70.11 ± 3.54	1617.67 ± 61.03	62.89 ± 8.45	1516.00 ± 67.19	1258.56 ± 52.39	1980.89 ± 84.95

Note: “Arms forward horizontally (standing)” means the distance between the tip of the middle finger and the ground after the subjects extend their arms horizontally in front of them in the standing position. “Hands up straight (standing)” means the distance between the tip of the middle finger and the ground after the subjects raise their hands vertically up to the highest position.

**Table 2 sensors-23-03850-t002:** Electrode location.

Muscle	Location
Pectoralis Major	Electrodes need to be placed along the anterior axillary fold away from the chest wall.
Biceps Brachii	Electrodes need to be placed on the line between the medial acromion and the fossa cubit at 1/3 from the fossa cubit.
Anterior Deltoid	Electrodes need to be placed at one finger width distal and anterior to the acromion.
Triceps Brachii	Electrodes need to be places at 1/2 on the line between the posterior crista of the acromion and the olecranon at 2 finger widths medial to the line.
Brachioradialis	Electrodes need to directly overlay the proximal portion of the muscle just distal to the elbow joint.
Upper Trapezius	Electrodes need to be placed at 1/2 on the line from the acromion to the spine on vertebra C7.

**Table 3 sensors-23-03850-t003:** Subjective evaluation of comfort level.

Score	−3	−2	−1	0	1	2	3
Comfort Level	Strongly uncomfortable	Uncomfortable	Less uncomfortable	Neutral	Comfortable	More comfortable	Most comfortable

**Table 4 sensors-23-03850-t004:** Correlation of sEMG indexes and test heights.

		PM	BB	AD	TB	BR	UT
RMS	Pearson correlation coefficients	0.726 *	0.845 **	0.986 **	0.953 **	0.812 *	0.976 **
	Sig.	0.042	0.008	0.000	0.000	0.014	0.000
iEMG	Pearson correlation coefficients	0.712*	0.835 **	0.977 **	0.950 **	0.882 **	0.966 **
	Sig.	0.048	0.010	0.000	0.000	0.004	0.000
MF	Pearson correlation coefficients	−0.882 **	0.815 *	−0.129	0.937 **	0.975 **	0.991 **
	Sig.	0.004	0.014	0.760	0.001	0.000	0.000

Note: * represents *p* < 0.05; ** represents *p* < 0.01.

**Table 5 sensors-23-03850-t005:** Significant differences in RMS and MF of different muscles.

sEMG Indicator	Group	PB	BB	AD	TB	BR	UT	F	*p*
RMS	G150	0.0775 ± 0.0084	0.1145 ± 0.0254	0.2365 ± 0.0726 ^ab^	0.0701 ± 0.0151 ^bc^	0.1741 ± 0.0201 ^abd^	0.1875 ± 0.0336 ^abd^	48.245	0.000
	G160	0.0801 ± 0.0158	0.0862 ± 0.0120	0.1593 ± 0.0575 ^ab^	0.0790 ± 0.0165 ^c^	0.1248 ± 0.0123 ^abd^	0.1553 ± 0.0299 ^abde^	39.386	0.000
	G170	0.0976 ± 0.0220	0.0665 ± 0.0084	0.1256 ± 0.0426	0.0528 ± 0.0081 ^ac^	0.0928 ± 0.0059 ^bd^	0.1266 ± 0.0212 ^bde^	35.436	0.000
MF	G150	15.3125 ± 2.3820	37.5313 ± 5.1763 ^a^	67.1250 ± 2.4312 ^ab^	24.0938 ± 10.0897 ^c^	31.6875 ± 3.6882 ^ac^	57.0313 ± 3.2989 ^abcde^	113.965	0.000
	G160	13.5938 ± 0.7514	36.5313 ± 4.7162 ^a^	44.1563 ± 3.4073 ^ab^	15.4562 ± 5.7590 ^bc^	43.6625 ± 5.5701 ^abd^	53.4563 ± 2.7044 ^abcde^	121.484	0.000
	G170	6.1719 ± 2.3773	33.1563 ± 3.8867 ^a^	56.3750 ± 5.4330 ^ab^	2.7969 ± 4.8564 ^bc^	34.9844 ± 7.3108 ^acd^	51.4375 ± 4.3829 ^abde^	163.394	0.000

Note: ^a^ indicates *p* < 0.05 compared with PB, ^b^ indicates *p* < 0.05 compared with BB, ^c^ indicates *p* < 0.05 compared with AD, ^d^ indicates *p* < 0.05 compared with TB and ^e^ indicates *p* < 0.05 compared with BR.

**Table 6 sensors-23-03850-t006:** Significant differences in RMS and MF of different groups.

sEMG Indicator	Muscle	G150	G160	G170	F	*p*
RMS	PM	0.0775 ± 0.0084	0.0801 ± 0.0045	0.0976 ± 0.0220	2.741	0.106
	BB	0.1145 ± 0.0254	0.0862 ± 0.0098 ^a^	0.0665 ± 0.0084 ^ab^	17.615	0.000
	AD	0.2365 ± 0.0726	0.1593 ± 0.0583 ^a^	0.1256 ± 0.0426 ^a^	7.399	0.004
	TB	0.0701 ± 0.0151	0.0790 ± 0.0157	0.0528 ± 0.0081 ^ab^	7.851	0.003
	BR	0.1741 ± 0.0201	0.1248 ± 0.0109 ^a^	0.0928 ± 0.0059 ^ab^	73.825	0.000
	UT	0.1875 ± 0.0336	0.1553 ± 0.0269 ^a^	0.1266 ± 0.0212 ^ab^	9.668	0.001
MF	PM	15.3125 ± 2.3820	13.5938 ± 0.7514	6.1719 ± 2.3773 ^ab^	36.896	0.000
	BB	37.5313 ± 5.1763	36.5313 ± 4.7162	33.1563 ± 3.8867	1.966	0.165
	AD	67.1250 ± 2.4312	44.1563 ± 3.4073 ^a^	56.3750 ± 5.4330 ^ab^	115.561	0.000
	TB	24.0938 ± 10.0897	15.4562 ± 5.7590 ^a^	2.7969 ± 4.8564 ^ab^	17.368	0.000
	BR	31.6875 ± 3.6882	43.6625 ± 5.5701 ^a^	34.9844 ± 7.3108 ^b^	9.363	0.001
	UT	57.0313 ± 3.2989	53.4563 ± 2.7044	51.4375 ± 4.3829 ^a^	5.148	0.015

Note: ^a^ indicates *p* < 0.05 compared with G150, and ^b^ indicates *p* < 0.05 compared with G160.

**Table 7 sensors-23-03850-t007:** Correlation matrix among different test muscles.

		PM	BB	AD	TB	BR	UT
Correlation	PM	1.000	0.330	0.059	0.237	0.050	0.135
	BB	0.330	1.000	0.610	0.350	0.478	0.683
	AD	0.059	0.610	1.000	0.239	0.503	0.644
	TB	0.237	0.350	0.239	1.000	0.187	0.277
	BR	0.050	0.478	0.503	0.187	1.000	0.530
	UT	0.135	0.683	0.644	0.277	0.530	1.000
Significance	PM		0.000	0.243	0.002	0.277	0.054
	BB	0.000		0.000	0.000	0.000	0.000
	AD	0.243	0.000		0.002	0.000	0.000
	TB	0.002	0.000	0.002		0.012	0.000
	BR	0.277	0.000	0.000	0.012		0.000
	UT	0.054	0.000	0.000	0.000	0.000	

**Table 8 sensors-23-03850-t008:** Variance percentage and cumulative percentage of principal component eigenvalues.

	Initial Eigenvalue			Extraction Sums of Squared Loadings		
Components	Total	Percentage of Variance	Accumulation	Total	Percentage of Variance	Accumulation
1	2.945	49.082	49.082	2.945	49.082	49.082
2	1.123	18.712	67.794	1.123	18.712	67.794
3	0.746	12.4411	80.235	0.746	12.4411	80.235
4	0.541	9.015	89.251	0.541	9.015	89.251

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
