# Peer review of "Analysis of the Hanging Actions and Operating Heights of Storage Furniture Suitable for the Elderly"

_sensors, 2023, doi:10.3390/s23083850_

Round 1

Reviewer 1 Report

my comments are in the pdf file 

Reviewer 2 Report

Comments and Suggestions:
In order to match the hanging height of the wardrobe with the use requirements, this paper uses an objective and subjective method to evaluate the influence of the height of the elderly on the hanging behavior. The topic and research content are reasonable, the research methods and technical routes are reasonable and feasible, and the analysis of the experimental data and results is justified. It has certain theoretical guidance and practical reference significance for the design of wardrobe furniture suitable for the elderly.
Questions and Suggestions:
1. The existing keywords are not suitable, it is suggested to change to: the elderly; storage furniture; hanging action; operating height (or hanging height); body height; comfort.
2. The title of the paper is "operating height". The main purpose of this paper is to study the relationship between "body height" and "operating height" of the elderly. G150 (1500-1599 mm), G160 (1600-1699 mm) and G170 (1700-1799 mm), and the number of males and females in each group should also be clearly explained in “2.1 Subjects”.
3. In “2.4.1", there is “In this test, the hanging rail height of the wardrobe was set to 8 levels, starting from 1536 mm and ending at 1984 mm. Every 64 mm was a test interval." It is better to change it to "In this test, the hanging rail height of the wardrobe was set to 8 levels, which were 1536 mm, 1600mm, 1664mm, 1728mm, 1792mm, 1856mm, 1920mm, and 1984 mm. Every 64mm was a test interval." In this way, it can correspond to the eight test heights or hanging heights shown in the lateral axis in FIG. 5.
4. For the six electrode locations shown in Figure 2: pectoralis major (PM), biceps brachii (BB), anterior deltoid (AD), triceps brachii (TB), brachialis radial (BR) and superior trapezius (UT), is there a specific size location (e.g., how far from the middle finger of the hand, etc.) every electrode location?
5. "B" shown in column 7 of Table 3 should be "BR".
6. The eight heights shown in the lateral axes in Figures 5, 6, 8, and 9 should be the “test height” or “hanging height” to distinguish them from the height of the subject.
